# Comparison of Landsat-8 and Sentinel-2 Data for Estimation of Leaf Area Index in Temperate Forests

**Lorenz Hans Meyer [1,\*], Marco Heurich [2,3], Burkhard Beudert [2], Joseph Premier [2] and Dirk Pflugmacher [1]**

[1] Geography Department, Humboldt Universität zu Berlin Unter den Linden 6, 10099 Berlin, Germany; dirk.pflugmacher@geo.hu-berlin.de

[2] Bavarian Forest National Park, Department of Visitor Management and National Park Monitoring Freyunger Str. 2, D-94481 Grafenau, Germany; Marco.Heurich@npv-bw.bayern.de (M.H.); Burkhard.Beudert@npv-bw.bayern.de (B.B.); Joe.Premier@npv-bw.bayern.de (J.P.)

[3] Chair of Wildlife Ecology and Management, University of Freiburg, Tennenbacher Straße 4, D-79106 Freiburg, Germany

[4] Bavarian Forest National Park, Department Nature Conservation and Research, Freyunger Str. 2, D-94481 Grafenau, Germany

[\*] Correspondence: lorenz.meyer@alumni.hu-berlin.de; Tel.: +49-15120787803

**Abstract:** With the launch of the Sentinel-2 satellites, a European capacity has been created to ensure continuity of Landsat and SPOT observations. In contrast to previous sensors, Sentinel-2's multispectral imager (MSI) incorporates three additional spectral bands in the red-edge (RE) region, which are expected to improve the mapping of vegetation traits. The objective of this study was to compare Sentinel-2 MSI and Landsat-8 OLI data for the estimation of leaf area index (LAI) in temperate, deciduous broadleaf forests. We used hemispherical photography to estimate effective LAI at 36 field plots. We then built and compared simple and multiple linear regression models between field-based LAI and spectral bands and vegetation indices derived from Landsat-8 and Sentinel-2, respectively. Our main findings are that Sentinel-2 predicts LAI with comparable accuracy to Landsat-8. The best Landsat-8 models predicted LAI with a root-mean-square error (RMSE) of 0.877, and the best Sentinel-2 model achieved an RMSE of 0.879. In addition, Sentinel-2's RE bands and RE-based indices did not improve LAI prediction. Thirdly, LAI models showed a high sensitivity to understory vegetation when tree cover was sparse. According to our findings, Sentinel-2 is capable of delivering data continuity at high temporal resolution.

**Keywords:** leaf area index; Sentinel-2; Landsat-8; vegetation; broadleaf forest; hemispherical photography

## 1. Introduction

Most of a plant's atmospheric exchanges happen through the leaves [1]. Therefore, the amount of leaf area is related to many plant–atmosphere processes, such as photosynthesis [2], evaporation and transpiration [3], rainfall interception [4], and carbon flux [5]. The quantity of that exchange is strongly correlated with the total amount of leaf area [1]. The leaf area of plant canopies is usually measured or estimated in the form of leaf area index (LAI), which is defined in broadleaf forests as the one-sided total green leaf area per unit ground surface area ($m^2/m^2$) [6]. LAI is considered by the United Nations Framework Convention on Climate Change (UNFCCC) and the Intergovernmental Panel on Climate Change (IPCC) as an essential climate and biodiversity variable [7]. However, measuring LAI at local to global scales is still an active area of research (e.g., [8,9]).

Multispectral satellite sensors have the ability to provide LAI estimates at varying temporal and spatial scales [10]. The most prominent LAI products are probably global-scale maps of LAI derived from coarse resolution sensors (≥100 m) such as MODIS, PROBA-V, and Sentinel-3, which in combination can provide long-term trends in LAI over nearly 20 years. Furthermore, the daily and near-daily temporal resolution (24–48 h) of these sensors makes them well-suited to observe LAI changes associated with phenological changes. Moderate-resolution sensors such as Landsat, with a pixel size between 10 and 100 m, have also been used to estimate LAI in the past, but due to their historically limited coverage, their relevance has been limited to local-scale analyses covering few Landsat scenes [11]. However, over the last few years, the availability of moderate-resolution sensor data has increased such that regional and global observations of many land surface properties have become feasible [12,13].

In June 2015 and March 2017, the European Space Agency (ESA) launched two Sentinel-2 (S-2) satellites to create a European capacity for operational Earth monitoring. The idea was to enable data continuity from preceding satellites such as the Landsat or the SPOT (Satellite Pour l'Observation de la Terre) satellites [14]. The Sentinel-2 mission offers an unprecedented combination of global coverage of land surfaces and a high temporal resolution, with a revisit time of 5 days. Sentinel-2 therefore includes six land monitoring bands that are comparable to Landsat-8 (L-8), but also includes three additional bands covering the red-edge (RE) spectrum [14]. The red-edge bands are centered at 704, 740, and 782 nm and have a bandwidth of 15, 15, and 20 nm, respectively. The RE is the prominent spectral feature of vegetation located between the red absorption maximum (680 nm) and the high reflectance in the Near-Infrared (750 nm) [15]. Furthermore, Sentinel-2's land surface bands have a spatial resolution of 10 m and 20 m compared to Landsat-8's 30 m. Because of these sensor differences, it would be interesting to understand which sensor performs best for predicting LAI.

The RE bands are well-positioned to improve the estimation of biophysical and biochemical vegetation parameters (e.g., [16–18]). However, there is no clear consensus in the literature as to whether RE bands and indices improve the retrieval of LAI. Some studies demonstrated a good correlation with LAI using hyperspectral data [16,19,20], while others reported contradictory results [21–23]. For example, Lee et al. (2004, [19]) showed that the spectral bands in the red-edge from the hyperspectral sensor AVIRIS were generally more important than those in the near-IR for predicting LAI across different biomes (including forests), but they also pointed out the importance of narrow wavelengths in hyperspectral sensors to adequately detect the red-edge. Conversely, Broge and Leblanc (2001) concluded that narrowband RE indices are not necessarily better predictors of LAI than the classic broadband indices. Few studies have tested the value of Sentinel-2's RE bands for predicting LAI in forest environments. Korhonen et al. (2017) compared Landsat-8's and Sentinel-2's abilities to estimate the forest LAI in a flat, boreal coniferous forest in Finland [24]. In this forest type, Sentinel-2 performed marginally better (0.6% to 2.1%) than Landsat-8, mainly due to the 705-nm RE band. Considering the inconsistent literature on the merit of red-edge bands for retrieving forest LAI, further studies on this topic are necessary.

Vegetation indices (VIs) are widely used to analyze vegetation changes over seasonal, interannual, and decadal time scales (e.g., [18,25]). VIs are numerical indices that are designed to maximize sensitivity to the vegetation characteristics while reducing confounding factors such as soil background reflectance or atmospheric effects [26]. For example, the normalized difference vegetation index (NDVI) is a well-established VI that correlates with the amount of photosynthetically active plant material [27]. Dash and Curran (2004) developed the first RE-based index to estimate chlorophyll content over wide spatial extents: the MERIS terrestrial chlorophyll index (MTCI) [28]. More recently, studies have used Sentinel-2 data to develop and test new RE-based indices, such as the inverted RE chlorophyll index (IRECI) and the Sentinel-2 RE position (S2REP) [15]. Prelaunch tests have shown promising results for estimating LAI of various crop types [15,29,30]. However, how well these indices correlate with LAI in broadleaf temperate forests remains to be tested.

The shrub layer of a forest can have a significant influence on remote-sensing-based LAI estimates [31]. The effect of shrub vegetation is particularly large in sparse forest plots. Although it is possible to differentiate between the understory and the actual trees to a certain degree in coniferous forests, this becomes impossible when the type of leaves in the tree layer are similar to the understory's [32]. In our study, the problem is acknowledged and addressed accordingly.

The objective of this study was to compare Landsat-8 and Sentinel-2 data for estimating LAI in temperate broadleaf forests. Specifically, we wanted to address the following research questions:

1.　Are empirical LAI models derived from Sentinel-2 MSI data more accurate than LAI models from Landsat-8 data?
2.　How well do RE bands and RE-based indices correlate with LAI compared to other vegetation indices?
3.　How does shrub vegetation influence empirical LAI models?
4.　How does the size of the pixel window matching the in situ plots influence empirical LAI models?

## 2. Study Area

The study area is located in the Bavarian Forest National Park (BFNP) in the southeastern part of Germany. The park extends along the main ridge of the Bohemian Forest; in the east, it is bounded by the border of the Czech Republic and the Sumava National Park (Figure 1). Since its establishment in 1970 and expansion in 1997, which almost doubled its size (from 13.229 hectares to 24.250 hectares), the BFNP has been Germany's oldest and also Bavaria's largest national park. Together with the Sumava National Park, it forms the largest strictly protected forested area in Central Europe.

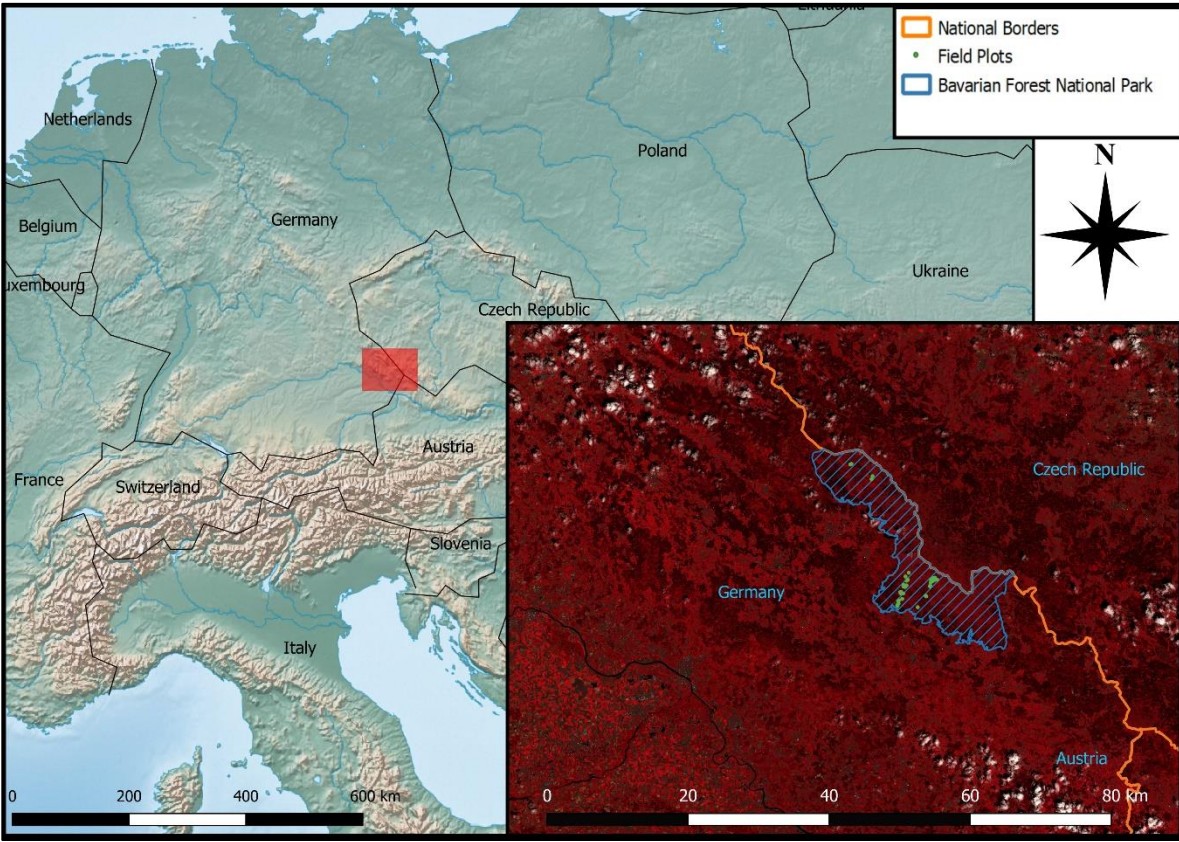

**Figure 1.** The Bavarian Forest National Park lies at the Czech–German border. The satellite picture (inset at the red rectangle) is a false-color composite of the NIR, red, and green bands of the Landsat-8 scene.

The region lies between the Central European and Continental Climate Zone (between Cfb and DfB in the Koeppen–Geiger climate classification). The climate of the region is diverse and depends much on the elevation. Average annual temperature ranges from 7.4 °C (804 mabove sea level.) to 3.5 °C (1436 m a.s.l.) in the subalpine areas. Precipitation varies strongly and lies between 810 mm per annum and 1766 mm [33]. Due to this relatively favorable climate, mixed montane forests occur. These consist mainly of Norway spruce (*Picea abies*), silver fir (*Abies alba*), European beech (*Fagus sylvatica*), and sycamore maple (*Acer pseudoplatanus*) [30]. In the lower ranges (around 600 m a.s.l.), the main forest type is spruce wetland forest with Norway spruce as the dominant tree species, with some silver birch (*Betula pendula*) and downy birch (*Betula pubescens*) [34]. Due to bark beetle (*Ips typographus)* outbreaks from the mid-1990s on, the cumulative area of killed mature spruce stands amounted to around 7088 ha as of 2018 [35].

## 3. Material and Methods

To answer the research questions, we selected near-cloud-free Landsat-8 and Sentinel-2 images acquired on the same day. As reference LAI measurements, we used estimates of effective LAI based on hemispherical photographs. We then developed and tested simple and multiple linear regression models using in situ LAI measurements as response and the sensors' spectral bands and vegetation indices as predictor variables. Finally, we assessed the influence of the pixel window for extracting spectral data corresponding to the field plots and the influence of shrub vegetation on the prediction accuracy of the models.

### *3.1. In Situ Measurements*

Field data were collected in August 2017 at 36 European beech (*Fagus sylvatica*) plots. Plots were selected within four 300-m-wide transects that followed the altitudinal gradient, varying slopes, tree species, and stand age. To span a large range of LAI, we selected plots with different tree densities (Table S1, Supplementary Material). Plots centers were marked and measured with a Leica Viva GS14 differential GPS device for 45 min under leaf-off conditions in March 2017. The raw data were then processed using SAPOS (the Bavarian state survey office's online portal), and the points with the lowest accuracy were measured once more for 30 min. This resulted in a spatial accuracy of around 5–10 cm.

At each plot, we took hemispherical photographs at five measurement locations (center + N/E/S/W side of a circle with a radius of 5 m) using a Canon EOS 6D camera with a Sigma 17–35 mm f2.8-4 EX aspherical HSM fisheye objective. To retrieve consistent LAI estimates, we followed the protocol suggested by Macfarlane (2007) [36]. The images were acquired close to sunrise or sunset under uniform overcast sky conditions to avoid the interference of direct sunlight, which can cause errors of up to 50% [37]. The resolution was set to maximum and the pictures were saved in the JPEG format. Aperture was set to minimum, with the camera in 'aperture-priority' mode. In an adjacent clearing, the exposure was measured and the shutter speed noted. Afterwards, the camera's mode was changed to 'manual' and the shutter speed lowered by two stops relative to that measured below open sky. At each photo location we took four images, totaling 20 images for each plot. The camera was put on a tripod with a height of approximately 1.3 m for stability.

The hemispherical photos were processed and analyzed with the freely available Gap Light Analyzer software [38], which derives the effective LAI ($LAI_e$) based on the gap fraction within canopies. $LAI_e$ assumes a random spatial distribution of the foliage and is defined for broad leaf canopies as follows [26]:

$$LAI_e = 2 \int_0^{\pi/2} \ln\left(\frac{1}{P(\theta)}\right) \cos\theta \sin\theta d\theta \tag{1}$$

To estimate gap fraction, we manually trained the Gap Light Analyzer to identify 'sky' and 'non-sky' pixels. Further input parameters were used to improve the result: (a) plot longitude/latitude, (b) elevation and slope, (c) orientation of the picture, and (d) time of picture acquisition. These parameters are used by the software to determine specific weighting factors (e.g., if the plot was at a

steep slope, the upward facing side appears to have less 'sky' pixels, and vice versa for the down-facing side). The mean LAI$_e$ value of the 20 images per plot was then calculated from the integral of the zenith angle range from 0 to 70° and taken as 'ground truth' for the further analyses. The measured values ranged from 1.52 to 5.58, with a median of 3.07, a mean value of 3.28' and a standard deviation of 1.15 (Figure 2).

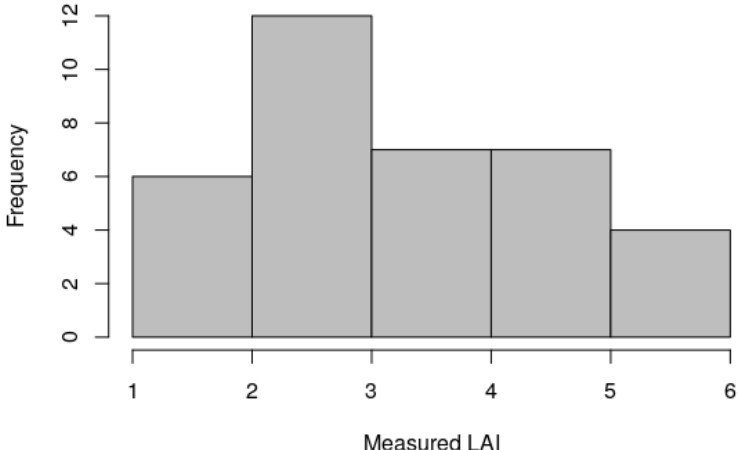

**Figure 2.** Distribution of in situ-measured effective leaf area index (LAI$_e$).

### 3.2. Landsat-8 and Sentinel-2 Data

We selected one scene of each sensor which showed no cloud cover above the study area. The Landsat-8 scene was obtained from the United States Geological Survey (USGS) Earth Explorer website (https://earthexplorer.usgs.gov); it was recorded at 09:57:22 CET on 13 July 2017, over path 192 and row 026. The Sentinel-2A MSI image was downloaded through the web portal of the Copernicus Project (https://scihub.copernicus.eu) and was recorded on the same day only 13 min after the Landsat-8 image, at 10:10:31 during the relative orbit 022 over tile T33UUQ. As the collection of the ground truth data started two weeks later, the scenes are within an acceptable time range as the canopy was already fully developed when the satellite images were recorded.

Atmospheric and topographic correction was applied using the Framework for Operational Radiometric Correction for Environmental Monitoring (FORCE) [39]. The atmospheric correction is based on radiative transfer theory using a combined image-, database-, and object-based estimation of aerosol optical depth [40]. The MODIS precipitable water product is used to characterize gaseous absorption [41]. For the topographic correction, FORCE uses a modified image-based C-correction with a 1-arc-second digital elevation model (DEM) derived from the shuttle radar topography mission (SRTM). From both sensors, the cirrus and aerosol bands were removed. A close examination by overlaying the images with a vector street map (obtained from http://openstreetmap.org) showed that both images were highly accurate in terms of geo-registration. For both sensors, we kept the native spatial resolution, i.e., 30 m for Landsat-8 and 10 m or 20 m for Sentinel-2.

Since many vegetation indices are highly correlated, we divided the indices and bands into five groups: near-infrared indices, atmospheric indices, RE indices, shortwave infrared indices, and spectral bands (Table S2, Supplementary Material). Near-infrared indices make use of near-infrared bands, often in relation to the red band. It is the largest group in this study, comprising 13 different indices, all of which were used in recent studies (e.g., [42,43]). Atmospheric indices use bands in the NIR and red spectrum, such as the near-infrared indices, but they also use the blue band to correct for aerosols [44,45]. For this paper, four different atmospheric indices were used. Red-edge indices use the RE area of the spectrum as the chlorophyll concentration contained in vegetation strongly correlates with the RE position (point of maximum slope around the red-edge). Thirteen indices were selected for this group [18,24,46]. Finally, shortwave infrared (SWIR) indices use the reflectance in the

shortwave infrared spectrum, which is sensitive to water content in vegetation and vegetation structure and can be related to LAI [47]. They are often used to detect drought stress and forest disturbances caused by wildfire, harvest, and insects [48]. The group consists of three different SWIR indices. All vegetation indices were calculated for Landsat-8 and for Sentinel-2, except the RE indices, which were only calculated for Sentinel-2.

To match the footprint of the ground measurements with the satellite pixel values, for each circular plot, we extracted the intersecting pixel values and the fraction of the intersecting pixels. Based on this, we calculated the area-weighted average at the native spatial resolution of each sensor. Linking hemispherical photo measurements with satellite data is particularly challenging because the hemispherical field of view also depends on canopy height. Several studies have reported that increasing the pixel extraction radius improved the modelling outcome [24,49,50], and therefore, the extraction radius was increased from 10 m to 40 m and 60 m in diameter. The resulting data contained the 36 plots with the reflectance data of each band (10 for Sentinel-2, 6 for Landsat-8).

*3.3. Statistical Analyses*

We used simple and multiple linear regression to develop $LAI_e$ prediction models. Linear regression aims to explain the distribution of a response variable by means of a linear relation of one or more predictor variables. For this study, $LAI_e$ estimated through the hemispherical photographs was the response variable and the spectral bands and vegetation indices were the predictor variables.

To evaluate the developed models, we calculated the root-mean-square error (RMSE) based on leave-one-out cross-validation (LOOCV). LOOCV is a special case of k-fold cross-validation. It systematically omits one plot at a time and yields an independent estimate of $LAI_e$ for each plot using the remaining n − 1 plots for developing the model. This way, each observation is used exactly once for validation and all data are used for training and validation [51].

Stepwise model selection was applied to the spectral bands and selected vegetation indices to identify the most parsimonious Sentinel-2 and Landsat-8 model. As the variable selection criterion, we used the Akaike information criterion through the 'stepAIC' function implemented in the 'MASS' package in R [52].

In addition to the multiple linear regression models, we also tested the predictive power of individual spectral bands and vegetation indices using simple linear regression. The indices and bands with the smallest RMSE of each group were selected for further analysis and reporting.

*3.4. The Effect of Shrub Vegetation*

Understory vegetation can influence satellite-estimated forest $LAI_e$ [31,53]. Since the hemispherical photos were taken 1.3 m above ground, understory vegetation such as grasses, ferns, mosses, and shrubs (mainly blackberry, *Rubus fruticosus L.)* lower than 1.3 m in height were not captured in our field-based $LAI_e$ estimates. However, low shrub vegetation was present at several plots with low tree density in our study area. To assess the effect of shrub vegetation on the $LAI_e$ models, we developed new $LAI_e$ prediction models by excluding plots with sparse tree cover and with shrub cover >50%. In total, six plots were excluded in this step.

## 4. Results

*4.1. Comparison of Landsat-8 and Sentinel-2 for Predicting LAI*

Sentinel-2 models performed slightly better than Landsat-8 models in predicting $LAI_e$. The best Sentinel-2 model based on stepwise selection explained 59.3% of the variance in $LAI_e$ and had an RMSE of 0.888, whereas the best Landsat-8 model explained only 48.7% of the variance, but had a lower RMSE of 0.813 (Figure 3). The Sentinel-2 model consisted of five bands including two RE bands: blue, green, RE1, RE3, and SWIR2. When excluding the Sentinel-2 RE bands from stepwise selection, the blue, green, NIR1, and SWIR2 bands were selected. The resulting model had a similar prediction

accuracy and goodness-of-fit (RMSE = 0.889, $R^2$ = 58.9%), indicating that including the red-edge bands did not improve LAI prediction accuracy. The Landsat-8 model consisted of three bands: green, NIR, and SWIR2.

Combining the best performing vegetation indices from each index group into multiple linear regression models also did not result in better fits than the simple linear models (see Section 4.2). The Landsat-8 indices used as predictors in multiple linear regression models were: pigment-specific simple ratio (PSSR), green atmospherically resistant vegetation index (GARI), and normalized burn ratio (NBR). The same variables were selected as predictors in the Seninel-2 model as well as the RE index PSRI. For both, Landsat-8 and Sentinel-2, stepwise selection yielded a single-variable model based on the GARI. Hence, the model statistics were almost identical with Sentinel-2 ($R^2$ = 0.450; RMSE = 0.887) and Landsat-8 ($R^2$ = 0.445; RMSE = 0.884).

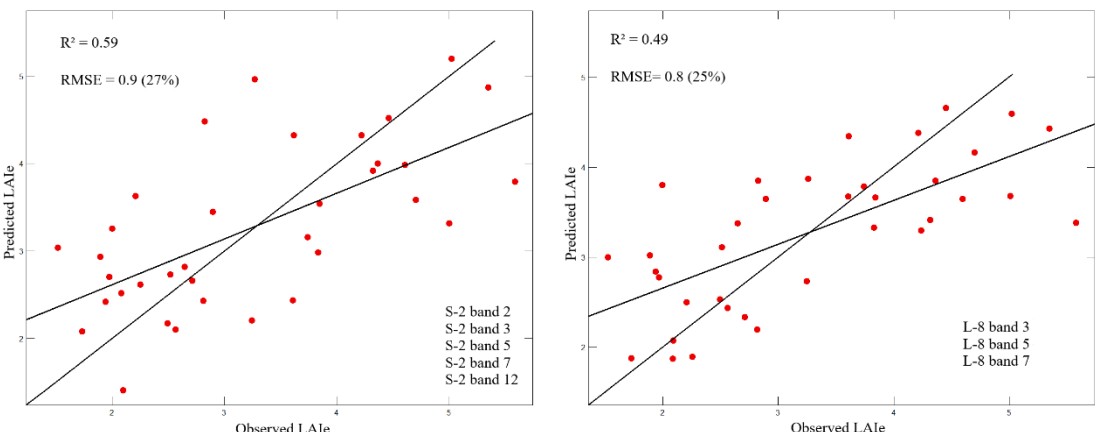

**Figure 3.** Observed vs. predicted $LAI_e$ of the best Sentinel-2 (S-2, right) and Landsat-8 (L-8, left) multiple linear regression models. RMSE: root-mean-square error.

### 4.2. Comparison of Single Bands and Vegetation Indices for Predicting $LAI_e$

Simple linear regression models based on RE indices and bands performed well, but not better than models based on other indices and bands. For Sentinel-2, the RE band RE3 performed best among the spectral bands, with an $R^2$ of 0.328 and an RMSE of 0.978, but this was only slightly better than the NIR band (8a) (RMSE = 0.993, $R^2$ = 0.306). In the absence of an RE band, the best Landsat-8 band was the NIR band (RMSE = 0.964, $R^2$ = 0.336), followed by the red band ($R^2$ = 0.283).

Overall, RE indices performed well as a group, but not better than the NIR and atmospheric indices (Figure 4). For example, the best index without an RE band (Landsat-8 PSSR) predicted $LAI_e$ with comparable accuracy (RMSE = 0.877). The RMSE was only 0.022 (or 2.45%), lower than the best RE indices. The two best RE indices were the plant senescence reflectance index (PSRI) ($R^2$ = 0.435, RMSE = 0.899) and the recently developed inverted red-edge chlorophyll index (IRECI) ($R^2$ = 0.403, RMSE = 0.919) [15]. The lowest accuracy was associated with the transformed chlorophyll absorption in reflectance index (TCARI) which uses the green, red, and RE1 bands ($R^2$ = 0.095).

Sentinel-2 NIR indices achieved $R^2$ values ranging from 0.354 to 0.454 (0.359 to 0.453 for Landsat-8, L-8). In the case of L-8, the NIR indices had the highest mean $R^2$ of all groups and the smallest mean RMSE (0.911). The PSSR index led to the best results of all univariate models for both sensors in this study, with an $R^2$ of 0.454 (Sentinel-2, S2) and 0.453 (L-8) (RMSE: 0.879 and 0.877, respectively) (Table 1). The normalized difference vegetation index (NDVI), as a widely used index for different applications, had average performance among NIR indices, with Landsat-8 being slightly better than Sentinel-2 ($R^2$: L-8: 0.420, S2: 0.411). The performance of the atmospheric indices as a group lay well in the range of the NIR indices. The best-performing index from this group was the GARI, with a resulting RMSE of 0.884 (for both S2 and L-8).

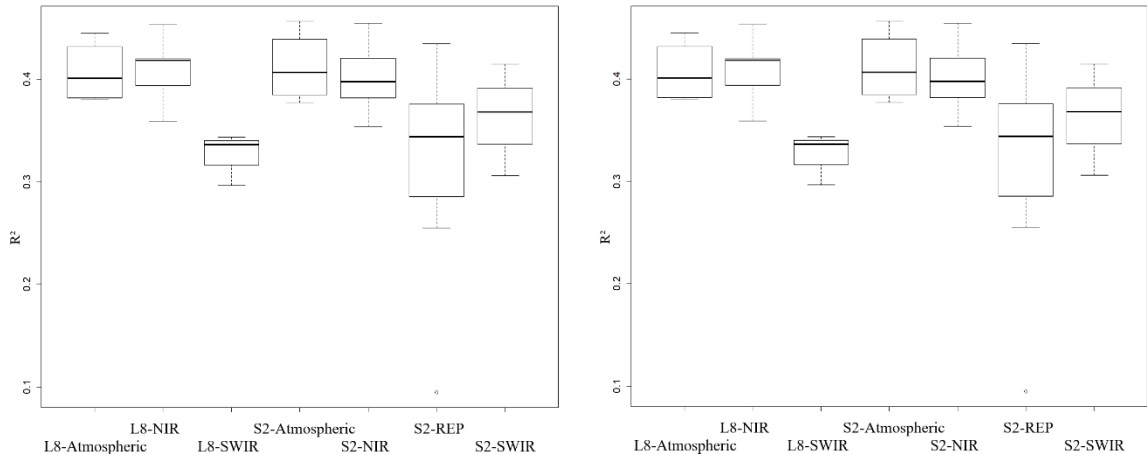

**Figure 4.** Boxplot of $R^2$ (left) and RMSE (right) of the different index groups. NIR: Near-Infrared; SWIR: shortwave infrared; REP: red-edge (RE) position.

**Table 1.** Model fit and RMSE of the best spectral bands and vegetation indices by index group for Sentinel-2 (S2) and Landsat-8 (L-8) at 60-m pixel window size. Best-performing indices are highlighted in bold.

| Sensor | Group | Index | RMSE | RMSE (%) | $R^2$ |
|--------|-------|-------|------|----------|-------|
| S2 | Bands | RE3 | 0.978 | 30.07 | 0.328 |
| S2 | **NIR** | **PSSR** | **0.879** | **27.02** | **0.454** |
| S2 | Atmospheric | GARI | 0.884 | 27.18 | 0.457 |
| S2 | REP | PSRI | 0.900 | 27.65 | 0.435 |
| S2 | SWIR | NBR | 0.918 | 28.21 | 0.414 |
| L-8 | Bands | NIR | 0.964 | 29.38 | 0.336 |
| L-8 | **NIR** | **PSSR** | **0.877** | **26.74** | **0.453** |
| L-8 | Atmospheric | GARI | 0.884 | 26.95 | 0.445 |
| L-8 | SWIR | NBR | 0.962 | 29.34 | 0.344 |

The SWIR indices showed the lowest performance compared to the NIR and atmospheric group, with an $R^2$ ranging from 0.306–0.414 (Sentinel-2) and 0.296–0.343 (Landsat-8). Only the NBR ($R^2$: S2: 0.414, L-8: 0.343) showed $R^2$ values which reached the level of the NIR and atmospheric indices.

*4.3. Model Sensitivity to Pixel Window Size*

Model RMSE decreased and $R^2$ increased systematically with increasing pixel window size, i.e., models based on the 60-m-diameter plot were consistently more accurate than models based on the 10-m-diameter plot (Figure 5). For Landsat-8, model $R^2$ was improved by between 2.8% (NBR) and 14.2% (spectral polygon vegetation index, SPVI) and RMSE decreased by between 0.021 (NBR) and 0.109 (SPVI). For Sentinel-2, model $R^2$ was improved by up to 20.2% (PSRI) but it even decreased for two RE indices (IRECI and wide dynamic range vegetation index red-edge, WDRVIre). Model RMSE for Sentinel-2 PSRI was improved by 0.158.

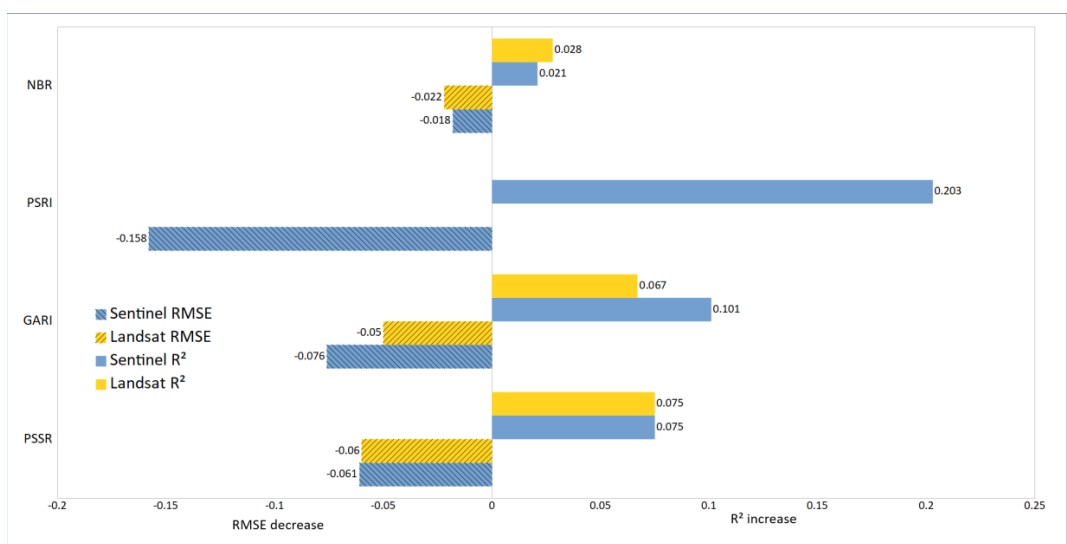

**Figure 5.** Effect of changing pixel window size from 10 m to 60 m on LAI$_e$ model RMSE and R$^2$ for the best-performing Landsat-8 and Sentinel-2 vegetation indices.

### 4.4. The Influence of the Shrub Layer on Model Sensitivity

Model fit and prediction accuracy generally improved further when plots with dense shrub vegetation were excluded in addition to the increase in pixel window size (Figure 6). The improvement was slightly larger on average for the Sentinel-2 models. The R$^2$ of the Sentinel-2 simple linear regression models was increased by between 1.6% (chlorophyll index green, Clg) and 27.1% (modified chlorophyll absorption in reflectance index, MCARI/MCARI2), and for the Landsat-8 models, was improved by between 1.5% (Clg) and 16.2% (visible atmospherically resistant index green, VARIg). The RMSE was also improved: it decreased by between 0.09 (Clg) and 0.21 (VARIg) (Landsat-8) and by between 0.04 (REP) and 0.28 (MCARI2) with Sentinel-2, respectively. Three of the Sentinel-2 RE indices reacted with a decrease in R$^2$ when removing shrub-covered plots, namely the S2REP (Sentinel-2 RE position, −8.0%) and the TCARI and the TCARI2 (transformed chlorophyll absorption ratio index; −5.6% and −0.2%, respectively). However, these indices were overall weakly correlated with LAI$_e$ (TCARI R$^2$: 0.094), suggesting that they were generally insensitive to variations in LAI$_e$.

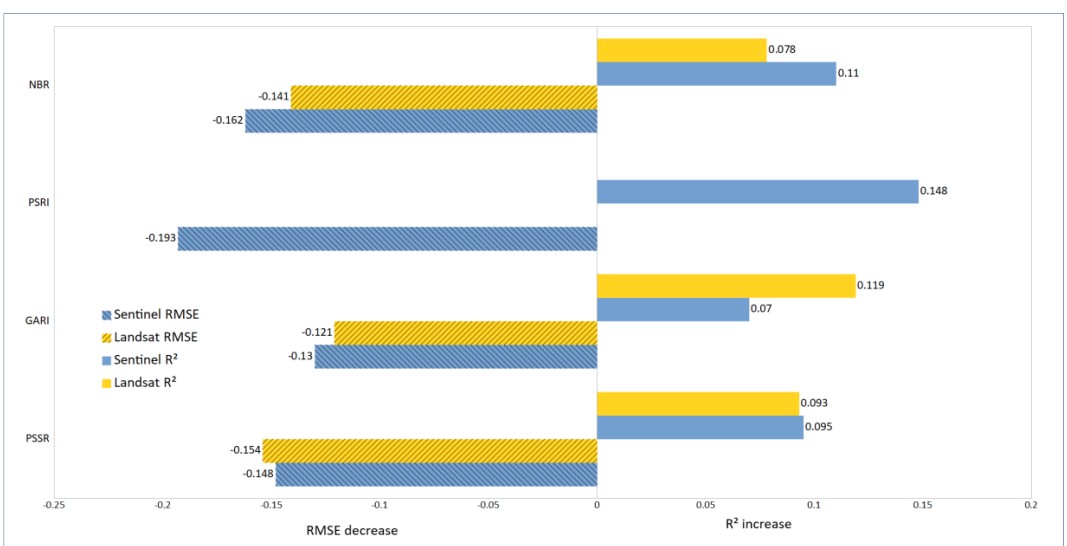

**Figure 6.** Effect of removing plots with a dense shrub layer on LAI$_e$ model RMSE and R$^2$ for the best-performing Landsat-8 and Sentinel-2 vegetation indices.

## 5. Discussion

The results of this study confirm that Sentinel-2 is capable of delivering data continuity for Landsat-8 at high temporal resolution. The differences in $R^2$ and RMSE between the vegetation indices used were marginal and not systematic. The RE bands and VIs are important, but brought no fundamentally better nor worse results compared to other indices. The shrub layer played a major role in the prediction of $LAI_e$, and model quality rose significantly when taking this effect into account.

### 5.1. Performance of Sentinel-2 and Landsat-8

The best prediction models were obtained using Landsat-8 data, although the difference to Sentinel-2 was small. Understandably, Sentinel-2's mission goal is to provide data continuity for Landsat (and other Earth-observation satellites such as SPOT), and therefore, most of its bands are compatible with Landsat. A correlation analysis of the individual bands confirmed strong agreements between the two sensors (Figure S1, Supplementary Material).

Comparing this study to other studies which estimated $LAI_e$ with satellite-based sensors shows that our model accuracies are at the lower end of the spectrum reported in the literature. Stenberg (2004) found a relation between the reduced simple ratio (RSR) and $LAI_e$, with an $R^2$ of 0.75 for homogeneous Scots pine plots [42]. Using only RE band 1 of Sentinel-2 MSI, Korhonen et al. (2017) reported an $R^2$ of 0.734 with an RMSE % of 19.6% for boreal forest canopies. Chen (2004) found an $R^2$ of 0.55 for $LAI_e$ measured through hemispherical photography and the IKONOS satellite sensor in Ponderosa pine forests [50]. One of the factors that may have influenced our results is the relatively complex forest structure and topography of the natural mountain forests studied.

Vegetation indices based on near-infrared bands were most strongly correlated with $LAI_e$, followed by atmospheric indices and RE indices. This suggests that near-infrared indices are sensitive for tracking changes in leaf area over time (e.g., for time series analysis), albeit they explained less than 50% of the variance in $LAI_e$. The results are different to Korhonen et al. (2017), who reported that near-infrared indices generally performed worse than RE indices for predicting LAI in boreal forests. In their study, the best NIR index (normalized ratio, NR) achieved an $R^2$ of 0.294, which is 15.9% lower than the best NIR index from our study (PSSR). Korhonen et al. (2017) used a relatively large reference sample to construct LAI models based on airborne lidar. Therefore, it is difficult to say whether the differences are site- or forest-type-specific or whether the source of reference data also played a role.

### 5.2. Importance of Sentinel-2's RE Bands for Estimating $LAI_e$

In this study, the performance of the RE bands was comparable to and not substantially better than the performance of other indices and bands. Although several prelaunch studies found a good linear relationship between chlorophyll content and RE indices ($R^2$ of 0.82 in Delegido's (2011) study and 0.88 in Frampton's (2013); both studies were conducted over agricultural sites in Spain) [15,18]. In our study, we were unable to reproduce such results for temperate broadleaf forests. There are several reasons which can help to explain these differences, one of them being that the indices used were developed and tested on agricultural sites and not on forests. Forest canopy is more complex, i.e., with stronger clumping, vertical differentiation, and illumination differences associated with canopy structure and gaps [54]. Also, while chlorophyll concentration (the total chlorophyll per unit area of ground) is commonly related to LAI, the correlation between both properties can be weak under certain conditions, e.g., under heat stress [55]. Therefore, if the relationship between $LAI_e$ and chlorophyll concentration breaks down in deciduous forests, chlorophyll-sensitive RE indices seem to have no advantages compared to NIR indices. Several studies show that $LAI_e$ and chlorophyll content are related, but also influence each other while predicting them from remote sensing data [56,57]. To increase the retrieval accuracy of chlorophyll content (or $LAI_e$), Xu et al. (2019) proposed pairing VIs within two-dimensional matrices in order to minimize confounding effects.

The performance of the three Sentinel-2 bands was variable. RE3, the band closest to the NIR spectrum, was the best predictor of LAI$_e$ among all Sentinel-2 bands, followed by RE2 and the two near-infrared bands (B8 and B8a). RE1, which is closest to the red spectrum, showed the lowest correlation with LAI$_e$. This is in sharp contrast to the findings of Korhonen et al. (2017), who found that RE1 was the best band (even outperforming the whole range of different vegetation indices) in predicting the LAI$_e$ for boreal forests [24]. The difference in performance of the RE bands in coniferous and broadleaf stands might well lie in the diverging reflectance of light of the two species. The different RE bands measure different parts of the RE, and indices often use one or more RE bands in the ratio with the reflectance in the green, red, or NIR area. As this ratio is considerably different (for example, the reflectance in the NIR is higher for broadleaf and lower in the green and red area than for coniferous vegetation), the same band or index can provide meaningful results for coniferous stands but perform rather poorly in broadleaf areas.

## 5.3. Influence of the Pixel Window Size on Model Accuracy

Landsat-8 and Sentinel-2 models improved when the size of the pixel extraction window was increased from 10 m in diameter to 60 m in diameter. The results are in line with the findings from other studies, which suggests that the effect of geolocation uncertainties in the satellite images and the vegetation-dependent field of view of the hemispherical cameras can be mediated to some degree [24,50,58]. Furthermore, atmospheric scattering from nearby pixels can also influence the reflectance spectra measured over the field plot area [24]. The effect may be more pronounced in heterogeneous landscapes. The level of heterogeneity in the study area may be higher than in managed, even-aged plantations because natural succession following disturbance is permitted. Sentinel-2 profited slightly more from increasing the plot size, which indicates that geolocation uncertainties had a stronger impact at the 10-m or 20-m resolution that at Landsat-8's 30-m resolution. This suggests that the Sentinel-2 models did not benefit from the higher spatial resolution; therefore, the higher spectral resolution of the 20-m Sentinel-2 bands seems more important than the higher spatial resolution of the 10-m bands.

## 5.4. Influence of Shrub Vegetation

Another drawback of measuring LAI in situ with hemispherical photography is that the camera is facing skywards, so the vegetation below the field of view is not recorded and therefore excluded from the ground truth. For dense plots with a closed canopy, the influence of the shrub layer is negligible [59]. However, under open canopies, shrubs influence the satellite-measured LAI strongly [60]. The spectral response (and true LAI) recorded by the satellite sensors was therefore similar to high-LAI plots, which explains the outliers and the relatively high RMSE in the lower segment of measured LAI in the models. It is possible to account for shrub vegetation to a certain degree in canopies where the leaf types in the lower layers and the overstory differ by using correction factors based on their spectral characteristics, but in the case of homogeneous broadleaf tree stands, their application was not possible [61]. To understand how much this effect influenced this study's findings, the plots with an open canopy but dense shrub vegetation were omitted, which led to significantly better results. The decrease in RMSE reached up to 28% in the case of several RE indices and it was systematic throughout all NIR, SWIR, and atmospheric indices. Excluding plots with understory vegetation substantially improved the simple linear regression models, i.e., $R^2$ values of several indices were 0.55 and greater, with the highest $R^2$ of the univariate linear models being 0.607 (VARIg). Indices which triangulate the relation between the green, red, and the NIR or RE shoulder seem to have improved most significantly. This applies to the MCARI, MCARI2, TVI, and MTVI of the group of the RE indices, which were improved by between 22.6% to 28.0%.

*5.5. Other Sources of Errors*

The measurement and estimation of LAI in the field is prone to errors on several levels. In this study, the method used for deriving LAI through hemispherical photography was in accordance with the procedure suggested by Macfarlane (2007) and other studies [62–64]. However, there are three commonly recognized discrepancies when measuring LAI through hemispherical photography. First, even with the best algorithms, LAI is derived from a two-dimensional picture and therefore it is difficult to quantify, especially in complex forest architectures. The saturation of optically measured LAI is usually reached at around LAI values of 5 [65]. Second, the estimation of LAI includes the contribution of woody elements, a so-called plant area index, rather than an actual leaf area index. Recent studies have shown that the error margin of LAI measured in a heterogeneous forest environment through hemispherical photography can reach −46.2% to +32.6% compared to LAI estimates derived with a terrestrial laser scanner [64]. Third, the photographic exposure affects the magnitude of canopy gap fraction, and therefore it is crucial to apply the right exposure in order to enable a clear distinction between sky and canopy pixels for later processing [66]. The right exposure is mainly dependent on the light conditions, and therefore the best case would be to have two cameras, one in an open field and one at the plot, to be able to adapt quickly to changing light conditions. As mentioned in the description of the study area, the weather conditions in the Bavarian Forest change very quickly, and therefore it can be assumed that the results would have improved if measurements were conducted with a simultaneous two-camera setup. Nevertheless, it is a widely used method, as it is fast, the equipment needed is relatively cheap to obtain, and it is an optical method, which means it has a certain similarity to the remotely sensed data.

## 6. Conclusions

Sentinel-2 is able to provide data continuity for existing Earth-observing missions such as the Landsat mission. The reported results lead to the conclusion that Sentinel-2 does not provide a systematically better (nor worse) estimate of LAI in a central European broadleaf forest, as the differences were marginal. Several studies showed that the LAI can generally be derived more precisely from the use of the RE area of a canopy's reflectance. However, in this case, it did not lead to better models than using well-established indices based on the NIR and red area of the spectrum.

Considering the relatively high RMSE, this study confirms Glenn's (2008) conclusion that VIs represent composite properties of LAI and canopy architecture and they are only moderately useful in predicting individual canopy properties [60]. The reported results indicate that Sentinel-2 profits more from averaging adjacent pixels and that the results improve by a large margin if the effect of the shrub layer is accounted for. Nevertheless, the high revisit frequency of the Sentinel-2 pair enables a better understanding of forest dynamics throughout the season.

Regarding the ground measurements, the established protocols are acceptable with two restrictions. The problem of saturation at LAI > 5 needs to be addressed, and researchers using hemispherical photography should be aware of this issue. The second restriction is the effect of the shrub layer. The results of our study suggest that the protocol by Macfarlane (2007) needs to be implemented in a way that accounts for the vegetation below measuring height, as this could lead to significantly better results, especially if the plots vary substantially in their specific structure [36]. Lidar data acquired with airborne scanners or in situ with terrestrial laser scanners can achieve a more detailed picture, as they overcome the problem of unaccounted-for understory, shrubs, clumped vegetation, and woody elements and enable the use of a significantly higher number of plots on which to base the modeling [64].

**Supplementary Materials:** The following are available online at http://www.mdpi.com/2072-4292/11/10/1160/s1: Figure S1: Correlation analysis of the corresponding bands, MD = mean difference. Table S1. Field plot characteristics, plots with dense shrub vegetation: 1 = present, 0 = absent. N/ha = Number of trees per hectare. Plot coordinates (Northing and Easting) are in UTM Zone 33N projection based on a WGS-84 datum and spheroid. Table S2. Vegetation indices used as predictors of effective LAI in simple and multiple linear regression models.

**Author Contributions:** Conceptualization: D.P., M.H., B.B. and L.H.M.; Methodology, D.P., M.H. and L.H.M., J.P. and B.B.; Resources, B.B., J.P. and M.H.; Writing—original draft preparation, L.H.M.; Writing—review and editing, M.H. and D.P.; Supervision, M.H. and D.P.

**Funding:** The research was funded by the administration of the Bavarian Forest National Park.

**Acknowledgments:** We would like to thank Amani Lwila, Benedikt Hutter, and Petra Stutz for helping with the field data collection, and David Frantz for helping with the processing of the Landsat-8 and Sentinel-2 data using FORCE. We also acknowledge the helpful suggestions of the anonymous reviewers.

**Conflicts of Interest:** The authors declare no conflict of interest.

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
