# Peer review of "Comparison of Landsat-8 and Sentinel-2 Data for Estimation of Leaf Area Index in Temperate Forests"

_remotesensing, doi:10.3390/rs11101160_

Round 1

Reviewer 1 Report

In the manuscript “Comparison between Landsat-8 and Sentinel-2 data for estimation of Leaf Area Index in temperate forests” authors aim to present the leaf area index (LAI) in the Bavarian Forest National Park using ground measurements and two sensors data. The study is interesting to learn the quality of two sensor data when predicting the LAIe. It was good to use Gap Light Analyse and other software to derive LAIe. The manuscript has a good flow and readable. I have a few comments to improve the contents of the manuscript.

There were some terms introduced as abbreviation without giving full name or given the full name later ( Eg. PSSR; L 242).

There were several spots having “Error! Reference source not found”. Is that broken link? Need to fix it.

The discussion is well written with relevant literature.

Three tables and a figure have given in the Appendix. Do these all be in the appendix? Generally, one appendix is OK. May be some results can be generated from the appendix table(s) (i.e.  Table 2)

NB: Minor comments were made in the .pdf using sticky notes

Author Response

Reviewer #1:   In the manuscript “Comparison between Landsat-8 and Sentinel-2 data for estimation of Leaf Area Index in temperate forests” authors aim to present the leaf area index (LAI) in the Bavarian Forest National Park using ground measurements and two sensors data. The study is interesting to learn the quality of two sensor data when predicting the LAIe. It was good to use Gap Light Analyse and other software to derive LAIe. The manuscript has a good flow and readable. I have a few comments to improve the contents of the manuscript.

Response: Thank you for your positive remarks.

Reviewer #1:   There were some terms introduced as abbreviation without giving full name or given the full name later (Eg. PSSR; L 242).

Response: We appreciate the comments regarding our inconsistent handling of abbreviations. We corrected this throughout the paper and wrote out the full names of the indices when they appeared the first time.

Reviewer #1:   There were several spots having “Error! Reference source not found”. Is that broken link? Need to fix it.

Response: We fixed the references to tables in figures in the revised manuscript.

Reviewer #1:   The discussion is well written with relevant literature.

Response: Thank you. We appreciate it.

Reviewer #1:  Three tables and a figure have given in the Appendix. Do these all be in the appendix? Generally, one appendix is OK. May be some results can be generated from the appendix table(s) (i.e.  Table 2)

Response: The appendix included two tables (one table across two pages) and one figure. All tables and figures provide additional information to support the analysis presented in the manuscript; they do not represent new data and/or analyses. The choice to put them into an Appendix (revised: Supplementary Material) was simply to improve the flow and readability of the manuscript, as they are not essential to understand and interpret the results. Table 1 lists the field plot measurements of LAIe that were used in the analysis and additional forest attributes, e.g, mean tree height, species. Table 2 provides a detailed overview of the vegetation indices that we used to predict LAIe, i.e., the equations we used and literature reference. We placed Table 2 into the Appendix, because it is very large to fit into the manuscript. In accordance with the journal guidelines, we moved the tables and figures into Supplemental Material.

Reviewer #1: Line 18. Please give a full name of RE.

Response: Thank you for this comment. We changed the first appearance of “RE” to “red-edge” (line 18), so the reader knows what the abbreviation stands for.

Reviewer #1: Line 61 Give full name of RE first and then abbreviation

Response: Thank you for pointing us to the missing full name of the abbreviation. We added the full name in the abstract (line 18) as well as the first time it appears in the full text (line 59)

Reviewer #1: Remove “Legend” from Map (line 98)

Response: Thank you for pointing us towards this issue. We removed the word “Legend” from the map.

Reviewer #1: Figure caption (line 101), wrong color composite should be changed to false-color composite

Response: We reworded the term to “false-color composite”

Reviewer #1: Line 109. Give scientific names in parenthesis.

Response: Thank you for pointing us to the missing scientific name of the Norway Spruce. We added the name in parenthesis and italic, like the others (line 127)

Reviewer #1: Line 122. Scientific names should be italic.

Response: We changed the font to italic (line 142).

Reviewer #1: line 242. PSSR abbreviation not written out.

Response: We appreciate the comments regarding our inconsistent handling of abbreviations. We corrected this throughout the paper and wrote out the full names of the indices when they appeared the first time.

Reviewer #1: line 255. GARI, the full name should be given when abbreviation is used the first time.

Response: We appreciate the comments regarding our inconsistent handling of abbreviations. We corrected this throughout the paper and wrote out the full names of the indices when they appeared the first time.

Reviewer #1: line 270, Figure 5. No Landsat 8 data for PSRI

Response: As the PSRI (Plant Senescence Reflectance Index) is an index using bands of the red-edge spectrum, this index was only calculated for Sentinel-2. We explain this in the manuscript line 210: „All vegetation indices were calculated for Landsat-8 and for Sentinel-2, except the RE indices were only calculated for Sentinel-2.“  

Reviewer #1: line 272, Figure 5 caption. Need to include keywords on Sentinel-2 and Landsat-8 data in Figure caption.

Response: We improved the figure caption as suggested by the reviewer as follows “Effect of changing pixel window size from 10 m to 60 m on LAIe model RMSE and R2 for the best performing Landsat-8 and Sentinel-2 vegetation indices”.

Reviewer #1: line 346. Remove hyphen

Response: We apologize for the inconsistency regarding the format of the units. We removed the hyphen between the number and the unit (line 403)

Reviewer #1: Table Plot Characteristics.   N/ha, N is not defined. Basal area, unit missing. Slope, use only one decimal. meter a.s.l. , case wrong.

Response: Thank you for your comments regarding the units of the table. We corrected it according to your suggestion.

Reviewer #1: Table Vegetation indices caption misleading, it is not clear what is meant with reference.

Response: We were referring to the peer-reviewed journal articles in which the indices and their formula were first reported. To avoid confusions, we removed the word “reference” and changed the table caption as follows: “Vegetation indices used as predictors of effective LAI in simple and multiple linear regression models.” Column header “Source” was renamed to “Reference”.

Reviewer 2 Report

Please find comments from the attachment below.

Author Response

Reviewer #2: This paper compares the performance of LAI estimation for deciduous forest from Landsat-8 and Sentinel-2 data. The topic of this paper is interesting and this paper can be a good reference for future LAI retrieval using Landsat-8 and Sentinel-2 images. However, this paper should clarify and discuss some issues. I can recommend it for publication after major revisions. Please, find my detailed comments below.

Response: Thank you for your positive feedback. We revised the text to improve the clarity and extended the discussion as you suggested.

Reviewer #2: I agree that only a few studies analyzed the merit of sentinel-2 RE bands for forest LAI retrievals. However, many studies tested the performance of RE bands from other sensors, such as RapidEye, Worldview and other hyperspectral instruments. Authors need to clarify the specific characteristics of RE in Sentinel-2, i.e., the number of RE bands, band width, center wavelength in discussion, which can avoid readers confusion.

Response: Thank you. We added more details about Sentinel-2's red-edge bands in the introduction (line 59), covering the wavelength at which the specific red-edge bands are centered as well as the bandwidth. (“The red-edge bands are centered at 704, 740 and 782nm and have a bandwidth of 15, 15 and 20nm respectively.” We also added additional references using other sensors (lines 67): “The RE bands are well positioned to improve the estimation of biophysical and biochemical variables such as LAI [18], chlorophyll content [19], vegetation cover, and biomass (e.g. [17,20,21]). Lee et al. (2004) showed that the spectral bands in the red-edge from the hyperspectral sensor AVIRIS were generally more important than those in the near-IR for predicting LAI across different biomes (including forests), but they also pointed out the importance of narrow wavelengths in hyperspectral sensors to adequately detect the red-edge.  Few studies have tested the value of Sentinel-2’s RE bands for predicting LAI in forest environments…”

Reviewer #2: I suggest authors add more discussions about performance of RE bands. The current version in Section 5.2 seems not enough. For example, authors can analyze the sensitivity of RE bands for LAI estimation. Some studies have demonstrated that at RE bands, chlorophyll content is more sensitive than LAI. This is a possible reason why RE indices is not substantially better than other indices. Authors can cite some papers to discuss these issues.

Response: Based on your valuable suggestions, we added a paragraph to broaden the discussion of Sentinel-2’s RE bands. The paragraph reads as follows (line 378):

“Also, while chlorophyll concentration (the total chlorophyll per unit area of ground) is commonly related to LAI, the correlation between both properties can be weak under certain conditions, e.g. under heat stress [52]. Therefore, if the relationship between LAIe and chlorophyll concentration brakes down in deciduous forests, chlorophyll-sensitive RE indices seem to have no advantages compared to NIR indices. Several studies show that LAIe and chlorophyll content is related but also influences each other while predicting it from remote sensing data [53,54]. To increase the retrieval accuracy of chlorophyll content (or LAIe), Xu et al. (2019) proposed a pairing of VI’s within 2-dimensional matrices in order to minimize confounding effects.“

Reviewer #2: Landsat-8 and Sentinel-2 data have different spatial resolutions. Authors should provide more details to deal with this issue. For example, the plot size, how to match the footprints between Landsat-8 and Sentinel-2 for the comparison of reflectances or LAI.

Response: To avoid additional sources of uncertainties related to resampling from 30m to 20m or vice versa, we kept the spatial resolution of Landsat-8 and Sentinel-2. To account for the different spatial resolutions when extracting pixel values, we calculated the weighted average of the pixels overlapped by the plot. We revised the paragraph to make this clearer:

Lines 212ff: “To match the footprint of the ground measurements with the satellite pixel values, we extracted for each circular plot the intersecting pixel values and the fraction of the intersecting pixel area. Based on this, we calculated the area weighted average at the native spatial resolution of each sensor.”

Reviewer #2: I am a little confused about the influence of understory vegetation in Section 5.4. Line 140 shows the height of camera is 1.3 meters and the camera faced upwards the vegetation. Authors should clarify that whether the shrub height is greater than 1.3 m. If yes, camera can also capture the shrub LAI. Authors should not exclude the plots with an open canopy but dense shrub vegetation.

Response:  Shrub vegetation in our study area is commonly below the 1.3m height threshold, and therefore below the field of view of the camera. To clarify this, we rephrased “understory vegetation” with “shrub vegetation”. The paragraph reads now as (line 239): Since the hemispherical photos were taken 1.3m above ground, understory vegetation such as grasses, ferns, mosses, and shrubs (mainly Blackberry, Rubus fruticosus L.) lower than 1.3m height were not captured in our field-based LAIe estimates. However, low shrub vegetation was present at several plots with low tree density in our study area.

Reviewer #2: Please correct the link error throughout the paper.

Response:  We corrected the link errors in the revised manuscript.

Reviewer #2: Line 224-225. Does that mean RE bands are more important than NIR for sentinel-2?

Response:  This is a good point. Stepwise selection only yields a single (best) model, which does not mean that there are no (best) alternative models. In the revised manuscript, we repeated step-wise selection by excluding the RE bands. The results show that the model without RE bands is equally strong as the model with RE bands. Hence, there is no evidence that suggests that RE bands are more important than NIR bands for Sentinel-2, which is confirmed by our simple linear regression models. To make this clear, we added the following information to the revised manuscript (line 255):

“When Excluding the Sentinel-2 RE bands from step-wise selection, bands x, y, z were selected. The resulting model had a similar prediction accuracy and goodness-of-fit (RMSE =0.889, R2=58.9%), indicating that including the red-edge bands did not improve LAI prediction accuracy.”

Further discussion about the importance of Sentinel-2’s RE bands is found in chapter 5.2 (line 369)

Reviewer #2: Line 281-283. Why R 2 decrease for three of Sentinel-2 RE indices? Can we say that these RE indices are not sensitive for understory?

Response:  Only indices that were overall weakly correlated with LAIe showed a decrease in accuracy when removing shrub plots. Hence the decrease was a function of the overall insensitivity to LAI not shrubs in particular. We revised the sentence to clarify this (line 329): “Three of the Sentinel-2 RE indices reacted with a decrease in R² when removing shrub covered plots, namely the S2REP (Sentinel-2 RE Position, -8.0%), the TCARI and the TCARI2 (Transformed Chlorophyll Absorption Ratio Index, -5.6% and -0.2%). However, these indices were overall weakly correlated with LAIe (TCARI R²: 0.094), suggesting that they were generally insensitive to variations in LAIe.”

Reviewer #2: 9. Table 2. Please add the longitude and latitude for each plot.

Response We added the plot coordinates in the revised table. To fit the layout requirements, we moved the whole appendix to the Supplementary Material.

Reviewer 3 Report

This is an interesting study which is of direct interest to the readers of Remote Sensing. I have made some suggestions below on how I think the manuscript as presented could be improved. But I am happy that the methodology is sound, and the results are founded and defensible. If the authors where happy to make the suggestions below to the manuscript  then I would support its immediate acceptance to Remote Sensing. 

Introduction

·     Generally well written with a broad range of literature reviewed.

·     Provides a good introduction to the study. 

·     Line 44/45. It would be good to give an indication of "coarse resolution” as this is quite an ambiguous term. 

·     Line 48. What time scales are we referring to here? You need to give specific details regarding these claims. 

·     Line 49. Define local scale. 

·     Line 50. Define medium resolution. 

·     Line 58. Need to define RE (Red-Edge).

·     Line 59. Define ‘slightly higher’. You need to avoid ambiguous terms such as these unless you offer specific details in brackets following. 

·     Line 61. I think you can provide more details here rather than say ‘expected to’. There is now enough published literature to suggest that the RE bands are important in vegetation monitoring in a range of environments. For example, I would recommend making reference to: https://www.mdpi.com/2073-4441/10/7/838which shows the relative importance of spatial/spectral resolution of Sentinel-2 data products. 

·     Line 69. I would make reference to how VI’s are based on the numerical relationship between data observed in different multispectral bands. 

·     Line 70. I would not say all VI’s are simple. 

Study Area: 

·     Section is detailed and complete. Some of this information might be better summarised in a table but this is a minor suggestion. 

Materials and methods: 

·     There appears to be some Word formatting errors for references from this point onwards in the paper. This needs to be fixed before acceptance. 

·     Section is detailed and gives a detailed account of the methodology. 

·     There are sections which could be written in a clearer style to aid reader understanding. 

·     Line 160. Why was only Sentinel 2-A data used and nothing from 2-B? 

·     Line 165. Please can you provide some more details on the correction methods used. 

·     Line 185. Was this a mathematically correct AW or a simple raster layer zonal statistic? 

·     Statistical analysis seems sound and appropriate.

Results: 

·     Quality/resolution of some of the graphics could be improved (but this is a minor point).

·     Results seem justified and defensible. 

Discussion: 

·     Line 295. Could you make reference to different significance levels here to avoid any ambiguity. 

·     Errors in referencing formats present again in this section. 

·     A detailed and generally well written results section. All findings are discussed in relation to relevant published literature. 

·     In Section 5.2 you focus on the importance of the red-edge bands. Is there something also to be said about the spatial resolution improvements of Sentinel data as well as these importance spectral improvements. For example while these 3 special RE bands have higher spectral resolution, they do have a coarser spatial resolution (when compared to Sentinel-2 band 8 which is also NIR). 

Author Response

Reviewer #3: This is an interesting study which is of direct interest to the readers of Remote Sensing. I have made some suggestions below on how I think the manuscript as presented could be improved. But I am happy that the methodology is sound, and the results are founded and defensible. If the authors where happy to make the suggestions below to the manuscript  then I would support its immediate acceptance to Remote Sensing. 

Response: Thank you for your positive review.

Reviewer #3: Study Area: Section is detailed and complete. Some of this information might be better summarized in a table but this is a minor suggestion.

Response: Regarding the briefness of that section, we decided that putting some of the climate/elevation/ forest type data into a table would decrease the readability of the whole text. To summarize the LAIe measurements, we thought about adding either a table or a figure but decided to put that information into an extra table which is found in the supplementary material (Table 1).

Reviewer #3: Materials and methods: There appears to be some Word formatting errors for references from this point onwards in the paper. This needs to be fixed before acceptance.

Response:  We corrected the link errors in the revised manuscript.

Reviewer #3: There are sections which could be written in a clearer style to aid reader understanding.

Response:  We carefully revised the manuscript to further improve clarity and readability.

Reviewer #3: Results: Quality/resolution of some of the graphics could be improved (but this is a minor point).

Response:  We increased the resolution of all graphs according to the journal guidelines, and improved the quality of the figures, e.g. increased font size, removed grid lines.

Reviewer #3: Line 44/45. It would be good to give an indication of "coarse resolution” as this is quite an ambiguous term.

Response: In the revised manuscript, we define coarse resolution sensors with a spatial resolution ≥ 250 m following Wulder et al., (2015)[12]. The revised sentence reads as follows (line 44): “The most prominent LAI products are probably global scale maps of LAI from coarse resolution sensors (≥ 250 m) such as MODIS, PROBA-V, and Sentinel-3, which together can now provide long-term trends in LAI for nearly 20 years.”

Reviewer #3: Line 48. What time scales are we referring to here? You need to give specific details regarding these claims.

Response: The revisit time of MODIS, PROBA-V and Sentinel-3 is between 24 and 48h. We added this information to the text (line 46): “. Furthermore, the daily and near-daily temporal resolution (24-48 hours) of these sensors makes them well suited to observe LAI changes associated with phenological changes.

Reviewer #3: Line 49. Define local scale.

Response: We revised the sentence as follows (lines 48): “Moderate resolution sensors like Landsat with a pixel size between 10-100 m have also been used to estimate LAI in the past but their relevance has been limited to local scale analyses covering few Landsat scenes [11].”

Reviewer #3: Line 50. Define medium resolution.

Response: We define moderate resolution between 10-100 m in accordance with Wulder et al. (2015). We revised the sentence as follows (lines 48): “Moderate resolution sensors like Landsat with a pixel size between 10-100 m have also been used to estimate LAI in the past but their relevance has been limited to local scale analyses covering few Landsat scenes [11].”

Reviewer #3: Line 58. Need to define RE (Red-Edge).

Response: We added a definition of the red-edge (line 59) as well as one for the red-edge position in line 60: “The RE is the prominent spectral feature of vegetation located between the red absorption maximum (680nm) and the high reflectance in the NIR (750nm) [14]. The red-edge position (point of maximum slope along the RE) as a quantification of the RE strongly correlates with the LAI as various studies have proved...”

Reviewer #3: Line 59. Define ‘slightly higher’. You need to avoid ambiguous terms such as these unless you offer specific details in brackets following.

Response: We removed the term and added the exact numbers of Sentinel-2's MSI (line 64): “Furthermore, Sentinel-2 has a spatial resolution of 10 to 20m compared to Landsat-8’s 30m.”

Reviewer #3: Line 61. I think you can provide more details here rather than say ‘expected to’. There is now enough published literature to suggest that the RE bands are important in vegetation monitoring in a range of environments. For example, I would recommend making reference to: https://www.mdpi.com/2073-4441/10/7/838which shows the relative importance of spatial/spectral resolution of Sentinel-2 data products.

Response: Thank you for suggesting this paper! We added reference to the suggested paper, changed “expected to” to “are well positioned for the estimation …" (line 67) and added one more reference to a paper where chlorophyll content is derived from Sentinel-2 red-edge bands. This should overall explain the importance of Sentinel-2's red-edge bands for remote-sensing of vegetation.

Reviewer #3: Line 69. I would make reference to how VI’s are based on the numerical relationship between data observed in different multispectral bands.

Response: We added as example a sentence about the NDVI (line 82): The Normalized Difference Vegetation Index (NDVI) is a well-established example for VIs, it is quantifying the characteristic of plants, which absorb light in the red and reflect to a high degree the near infrared spectrum of the light.

Reviewer #3: Line 70. I would not say all VI’s are simple.

Response: You are right, not all (used) VI’s are simple, we deleted the word, so the sentence is (line 80): “VIs are numerical indices that are designed to maximize sensitivity to the vegetation characteristics while reducing confounding factors like soil background reflectance or atmospheric effects.

Reviewer #3: Line 160. Why was only Sentinel 2-A data used and nothing from 2-B?

Response: Sentinel 2-A and 2-B carry the same instrument: MSI. Hence, we expect that our findings based on 2-A also apply to 2-B. There may be small calibration-related differences between the two sensors, but we expect them to be small relative to the differences between Landsat-8 and Sentinel-2. Chastain et al. (2019) did a systematic comparison of Landsat and Sentinel-2 sensors and found that the two MSI instruments were as similar as Landsat ETM+ and OLI. However, the study used top-of-atmosphere reflectance data, so we should expect that a good proportion of these differences is related to the atmosphere (see also Flood, 2017). Also, extensive cloud cover did not yield a cloud-free 2-B scene for the 2017 vegetation season in our study area.

Chastain, R., Housman, I., Goldstein, J., Finco, M., 2019. Empirical cross sensor comparison of Sentinel-2A and 2B MSI, Landsat-8 OLI, and Landsat-7 ETM+ top of atmosphere spectral characteristics over the conterminous United States. Remote Sens. Environ. 221, 274–285. https://doi.org/10.1016/j.rse.2018.11.012Flood, N., 2017. Comparing Sentinel-2A and Landsat 7 and 8 using surface reflectance over Australia. Remote Sens. 9, 1–14. https://doi.org/10.3390/rs9070659

Reviewer #3: Line 165. Please can you provide some more details on the correction methods used.

Response: We added more details how the FORCE framework corrects the raw data. The paragraph now reads (line 185): “The atmospheric correction is based on radiative transfer theory using a combined image-, database- and object-based estimation of aerosol optical depth [38]. The MODIS precipitable water product is used to characterize gaseous absorption [39]. For the topographic correction FORCE uses a modified image-based C-correction with a 1-arc-second digital elevation model (DEM) derived from the Shuttle Radar Topography Mission (SRTM).

Reviewer #3: Line 185. Was this a mathematically correct AW or a simple raster layer zonal statistic?

Response: We used mathematically correct averaged weighting. We added sentences to clarify that (line 213) To match the footprint of the ground measurements with the satellite pixel values, we extracted for each circular plot the intersecting pixel values and the fraction of the intersecting pixel. Based on this, we calculated the area weighted average at the native spatial resolution of each sensor"

 Reviewer #3: Line 295. Could you make reference to different significance levels here to avoid any ambiguity.

Response: It is difficult to make a statement about the significance of the observed differences. The differences in RMSE of ~0.1 was comparable to the differences found in a similar study in a boreal forest, although the performance of the sensors was reversed. More replication on a wider range of study sites is needed. We realize our statement was too speculative. Hence, revised the sentence as follows (lines 345):

Original: “The best prediction models were obtained using Landsat-8 data, although the difference to Sentinel-2 may be too small to be of scientific significance.”

Revised: “The best prediction models were obtained using Landsat-8 data, although the difference to Sentinel-2 was small.”

Reviewer #3: In Section 5.2 you focus on the importance of the red-edge bands. Is there something also to be said about the spatial resolution improvements of Sentinel data as well as these importance spectral improvements. For example while these 3 special RE bands have higher spectral resolution, they do have a coarser spatial resolution (when compared to Sentinel-2 band 8 which is also NIR).

Response: We added in section 5.3 a paragraph where we discuss this issue. We saw it more fitting in that section as we discuss there the effect of the window pixel size. The added paragraph reads as follows (line 409ff):

“Sentinel-2 profited slightly more from increasing the plot size which indicates that geolocation uncertainties had a stronger impact at the 10m or 20m resolution that at Landsat-8’s 30m resolution. This suggests, that the Sentinel-2 models did not benefit from the higher spatial resolution, therefore the higher spectral resolution of the 20m Sentinel-2 bands seem more important than the higher spatial resolution of the 10m bands.”

Round 2

Reviewer 2 Report

I read carefully the responses to comments as well as the revised version of the manuscript. The manuscript was significantly improved and all comments from reviewers were taken into account. I have no comments now. Thanks!

Author Response

Thank you very much! We highly appreciated your previous comments and are glad that you see no more issues regarding the quality of the paper.

Reviewer 3 Report

I would like to thank the authors for their detailed responses to my initial comments. I hope they feel as I do that this is a much stronger manuscript now. I do have some queries about their responses which I have noted below. If these could be addressed then I would immediately support this manuscripts acceptance to Remote Sensing.

I appreciate that you have tried to quantify different resolutions (e.g. coarse, medium, local etc.). However there are some inconsistencies in the classification of definitions that you offer in the revised manuscript. You define coarse resolution as >250m (I think this is a very fair definition). You then go on later in the paper and define medium resolution as 10-100m. So a fair question would be what comes between medium and coarse. I appreciate that this may seem minor, but the language when discussing resolution and its importance as a factor in this type of work must be clear. Some I think might define medium as up to 250m?

Line 59. I think this still could be clearer. You make reference to Sentinel 2 having a resolution of 10 to 20m, and Landsat 30m. This needs to be clear that you are talking about NIR bands – as all Sentinel bands range from 10-60m.

Line 67. The author’s response indicates that reference was made to a suggested paper. However I cannot see this in the reference list. If reference was made to the suggested paper then there is an error in the reference list as it is not included. This just needs to be checked! https://www.mdpi.com/2073-4441/10/7/838

Line 80. You say the NDVI ‘quantifies the characteristics of plants’. This is not strictly true; the NDVI can be used in detecting photosynthetically active plant material, from which plant stress can be inferred (https://www.mdpi.com/2073-4441/10/7/838). This is not quite the same as saying it quantifies the characteristics of plants.

Line 80. It would also be appropriate here to cite Tucker (1979) as the original author of NDVI. Tucker, C.J. (1979) 'Red and Photographic Infrared Linear Combinations for Monitoring Vegetation', Remote Sensing of Environment, 8(2),127-150.

I would also like to thank the authors for their kind acknowledgement to the reviewers at the close of the paper.

Author Response

Authors’ response to the reviewers comments:

Reviewer #3: I would like to thank the authors for their detailed responses to my initial comments. I hope they feel as I do that this is a much stronger manuscript now. I do have some queries about their responses which I have noted below. If these could be addressed then I would immediately support this manuscripts acceptance to Remote Sensing.

Authors: Thank you.

Reviewer #3: I appreciate that you have tried to quantify different resolutions (e.g. coarse, medium, local etc.). However there are some inconsistencies in the classification of definitions that you offer in the revised manuscript. You define coarse resolution as >250m (I think this is a very fair definition). You then go on later in the paper and define medium resolution as 10-100m. So a fair question would be what comes between medium and coarse. I appreciate that this may seem minor, but the language when discussing resolution and its importance as a factor in this type of work must be clear. Some I think might define medium as up to 250m?

Authors: In the first revision, we used the definition from Wulder et al. (2015) based on existing optical satellite sensor data. To accommodate the reviewer’s comment, we changed the definition of coarse resolution sensor data to > 100 m (line 51), which is in line with the definition by Joshi et al. (2016).

Joshi, N., M. Baumann, A. Ehammer, R. Fensholt, K. Grogan, P. Hostert, et al. 2016. A review of the application of optical and radar remote sensing data fusion to land use mapping and monitoring. Remote Sens. 8, 1–23.

Wulder, M.A., Hilker, T., White, J.C., Coops, N.C., Masek, J.G., Pflugmacher, D., Crevier, Y., 2015. Virtual constellations for global terrestrial monitoring. Remote Sens. Environ. 170, 62–76. https://doi.org/10.1016/j.rse.2015.09.001

Line 59. I think this still could be clearer. You make reference to Sentinel 2 having a resolution of 10 to 20m, and Landsat 30m. This needs to be clear that you are talking about NIR bands – as all Sentinel bands range from 10-60m.

Authors: We added text to make clear that we are referring to the terrestrial monitoring bands (and not the atmospheric bands). The revised sentence reads as follows (line 69): “Furthermore, Sentinel-2’s land surface bands have a spatial resolution of 10m and 20m compared to Landsat-8’s 30m”

Line 67. The author’s response indicates that reference was made to a suggested paper. However I cannot see this in the reference list. If reference was made to the suggested paper then there is an error in the reference list as it is not included. This just needs to be checked! https://www.mdpi.com/2073-4441/10/7/838

Authors: We apologize for that referencing error! A sentence summarizing the findings of West et al. (2018) (line71) was added: “Under extreme drought the RE variant of the Normalized Difference Vegetation Index (NDVI) outperformed  significantly the NIR variants of the NDVI [22].” and included the reference to their study in the list of references.

Line 80. You say the NDVI ‘quantifies the characteristics of plants’. This is not strictly true; the NDVI can be used in detecting photosynthetically active plant material, from which plant stress can be inferred (https://www.mdpi.com/2073-4441/10/7/838). This is not quite the same as saying it quantifies the characteristics of plants.

Authors: We revised the sentence according to the reviewer’s suggestion as follows (line 87): “For example, the Normalized Difference Vegetation Index (NDVI) is a well-established VI that correlates with the amount of photosynthetically active plant material.”

Line 80. It would also be appropriate here to cite Tucker (1979) as the original author of NDVI. Tucker, C.J. (1979) 'Red and Photographic Infrared Linear Combinations for Monitoring Vegetation', Remote Sensing of Environment, 8(2),127-150.

Authors: We added the suggested paper as reference to the NDVI (line 88).

I would also like to thank the authors for their kind acknowledgement to the reviewers at the close of the paper.
